# Field-Evolved *ΔG210-ppo2* from Palmer Amaranth Confers Pre-emergence Tolerance to PPO-Inhibitors in Rice and *Arabidopsis*

**DOI:** 10.3390/genes13061044

**Published:** 2022-06-10

**Authors:** Pamela Carvalho-Moore, Gulab Rangani, Ana Claudia Langaro, Vibha Srivastava, Aimone Porri, Steven J. Bowe, Jens Lerchl, Nilda Roma-Burgos

**Affiliations:** 1Department of Crop, Soil, and Environmental Sciences, University of Arkansas, Fayetteville, AR 72704, USA; pcarvalh@uark.edu (P.C.-M.); grangani@uark.edu (G.R.); vibhas@uark.edu (V.S.); 2Corteva Agriscience, Nova Mutum 78450, MT, Brazil; langaro.ac@gmail.com; 3BASF SE, RP 67056 Limburgerhof, Germany; aimone.porri@basf.com (A.P.); jens.lerchl@basf.com (J.L.); 4BASF Agricultural Solutions, Research Triangle Park, NC 27713, USA; steven.bowe@basf.com

**Keywords:** transgenic rice, *Amaranthus palmeri*, fomesafen, protoporphyrinogen oxidase, herbicide resistance, root growth inhibition

## Abstract

Resistance to protoporphyrinogen IX oxidase (PPO)-inhibitors in *Amaranthus palmeri* and *Amaranthus tuberculatus* is mainly contributed by mutations in the PPO enzyme, which renders herbicide molecules ineffective. The deletion of glycine210 (*ΔG210*) is the most predominant PPO mutation. *ΔG210-ppo2* is overexpressed in rice (*Oryza sativa* c. ‘Nipponbare’) and *Arabidopsis thaliana* (Col-0). A foliar assay was conducted on transgenic T_1_ rice plants with 2× dose of fomesafen (780 g ha^−1^), showing less injury than the non-transgenic (WT) plants. A soil-based assay conducted with T_2_ rice seeds confirmed tolerance to fomesafen applied pre-emergence. In agar medium, root growth of WT rice seedlings was inhibited >90% at 5 µM fomesafen, while root growth of T2 seedlings was inhibited by 50% at 45 µM fomesafen. The presence and expression of the transgene were confirmed in the T_2_ rice survivors of soil-applied fomesafen. A soil-based assay was also conducted with transgenic *A. thaliana* expressing *ΔG210-ppo2* which confirmed tolerance to the pre-emergence application of fomesafen and saflufenacil. The expression of *A. palmeri* *ΔG210-ppo2* successfully conferred tolerance to soil-applied fomesafen in rice and Arabidopsis. This mutant also confers cross-tolerance to saflufenacil in Arabidopsis. This trait could be introduced into high-value crops that lack chemical options for weed management.

## 1. Introduction

Palmer amaranth (*A. palmeri* L. Wats.) is a dioecious plant native to the desert region of the southwestern United States and northwest Mexico. Recent surveys denoted *A. palmeri* as one of the toughest weeds to manage in crop production in the southern US [1]. Since this weed has become resistant to several herbicide chemistries, its negative impact on crop production is high. Thus far, *A. palmeri* populations have evolved resistance to nine herbicide sites of action. These include inhibitors of acetolactate synthase, 5-enolpyruvyl-shikimate-3-phosphate synthase, microtubule assembly, photosystem II, 4-hydroxyphenylpyruvate dioxygenase, protoporphyrinogen oxidase, very-long-chain fatty acid synthesis, and glutamine synthetase as well as synthetic auxins [2,3]. Among the protoporphyrinogen oxidase (PPO, EC 1.3.3.4) inhibitors, fomesafen is highly used to control many dicot and monocot weeds in soybeans [4].

Herbicides that inhibit PPO control susceptible plants by stopping the oxidation of protoporphyrinogen IX into protoporphyrin [5]. This inhibition induces an excessive accumulation of the substrate (protoporphyrinogen IX), which leaks into the cytoplasm, where it reacts instantly with free oxygen. This oxidation process produces the highly photosensitive protoporphyrin IX, which generates singlet oxygen when exposed to light. Singlet oxygen molecules cause lipid peroxidation, cellular membrane disruption, the disintegration of cells, loss of carotenoids and chlorophyll (bleaching effect), and cell death [6,7,8,9,10]. In plants, PPO is nuclear-encoded in two forms: PPO1, which is compartmentalized in the chloroplast, and PPO2, which is compartmentalized in the mitochondria and, in a few species, also in the chloroplast [8,11,12]. In the *Amaranthus* genus, PPO2 is targeted to both mitochondria and plastids. The coding sequence of PPO2 has two in-frame start codons that result in two different lengths of PPO protein sequences. These two PPO proteins with different sequence lengths are thought to be targeted at the mitochondria and chloroplasts. Therefore, *Amaranthus* species are expected to have PPO2 activity in both compartments, which might be a requirement to confer herbicide resistance [13].

Fomesafen is one of the PPO-inhibiting herbicides with long residual soil activity, in addition to strong foliar activity. Consequently, it is applied preplant, pre-emergence, over-the-top, or postemergence-directed in various crops. Rice (*O. sativa* L.) does not have commercial tolerance to fomesafen, either applied pre-emergence or postemergence. In addition, rice is more sensitive to soil-applied than foliar-applied fomesafen. The half-life of fomesafen in soil varies from 8.5 to 100 days depending on the soil’s physical, chemical, and biological features and environmental conditions [3,14,15]. Evaluating the effect of residual fomesafen in soil on rotational crops, Cobucci et al. [16] recommended that rice not be planted for at least 95 days after fomesafen application. The label for fomesafen does not allow planting rice within 10 months from application; otherwise, the crop may be severely injured [17]. Resistance to fomesafen in *A. palmeri* is weaker when the herbicide is applied pre-emergence compared to postemergence, despite the presence of resistance-conferring *ppo2* mutations. Soil-applied fomesafen reduces the germination of fomesafen-resistant *A. palmeri* by 64% [18]. Saflufenacil, flumioxazin, and sulfentrazone are other examples of soil-applied PPO-inhibiting herbicides commonly used in soybeans across the USA. Saflufenacil is the newest member of this group and is still effective on PPO-resistant populations.

In Amaranthaceae, resistance to PPO inhibitors is mainly attributed to mutations in *PPO2*. The three base pair deletion resulting in the disappearance of glycine at position 210 (*ΔG210*) of PPO2 is the first mutation reported to confer resistance to PPO herbicides in *A. tuberculatus* [19]. Later, this same mutation was identified as the resistance mechanism in a PPO-resistant *A. palmeri* population [20]. The substitution of arginine with glycine or methionine at position 128 (*R128G* or *R128M*) also confers resistance [21]. The latest resistance-conferring mutation discovered in *A. palmeri* is a glycine substitution with alanine at position 399 (*G399A*) [22]. Follow-up surveys on *A. palmeri* populations from the mid-Southern US showed *ΔG210* as the predominant mutation along with the accumulation of other mutations in highly resistant populations [23,24]. The potential effect of these mutations on the functionality of PPO2, and hence resistance to herbicide, can be predicted using molecular modeling. A recent study by Noguera. et al., [23] using crystal structure molecular models of wild-type *A. tuberculatus* PPO2 and its *ΔG210* mutant showed that the deletion of G210 in *ppo2* causes the enlargement of the binding pocket or active site that allows water to enter more easily, which in turn interferes with the binding of fomesafen. This mutation causes minimum perturbation of substrate binding (or enzyme functionality), which explains the predominance of *ΔG210* among resistant populations.

One way to determine the contribution of a particular mutation to whole-plant resistance is by expressing it in a heterologous system. Researchers have shown that the presence of the above-cited mutations reduces the binding affinity of PPO herbicides to PPO2, consequently decreasing the herbicide effect on resistant weeds [19,25,26]. In this study, we characterized *ΔG210* mutation from *A. palmeri* using rice (*O. sativa* cv. ‘Nipponbare’) as a grass crop model and *A. thaliana* as a highly sensitive broadleaf plant model. A transgenic rice line was created using the *A. palmeri ppo2* gene, and its response to fomesafen was evaluated. Additionally, we evaluated the response of the transgenic Arabidopsis line to soil-applied fomesafen and saflufenacil (Appendix A).

## 2. Materials and Methods

### 2.1. Generation of Transgenic Rice

A transgenic rice line was generated under controlled conditions at the University of Arkansas in Fayetteville, Arkansas, USA. The full-length cDNA (Genbank accession number MF583746) of fomesafen-resistant *A. palmeri* plant from field population 15CLA-A [20] was obtained using the *ppo2* primer pair; kpnApxF: 5′-ggggtacccgggTAAACTGATCTTATGTTAATTC-3′ and sphApxR: 5′-ggaattcgagctcgcatgcTTACGCGGTCTTCTCATCCATC-3′ and RT-PCR procedure using 2 µL of the first-strand cDNA. This full-length cDNA also contained a dual organelle targeting signal. PCR amplification was performed using the following program: hot start at 95 °C for 2 min, denaturation at 95 °C for 1 min, annealing at 58 °C for 1 min, extension at 72 °C for 2 min for 39 cycles, followed by 72 °C for 10 min. The resulting PCR products were then gel-purified using the GeneClean kit (MP Biomedicals, Solon, OH, USA), digested with *XmaI* and *SacI* (New England Biolabs, Ipswich, MA, USA) and ligated with *XmaI*- and *SacI-*digested pRP7, resulting in the plasmids pACL1, which contained the mature protein coding region for *A. palmeri ppo2* isolated from the fomesafen-resistant plant under the control of maize ubiquitin (ZmUbi) promoter (Appendix A). To ensure the accuracy of the cloned *ppo2* sequence, PCR amplicon using pACL1 plasmid as a template was gel purified (GeneJET gel extraction kit, Thermo Fisher Scientific, Grand Island, NY, USA) and sequenced. The presence of intact *ppo2* genes with G210 mutation, henceforth referred to as *ΔG210-Apppo2*, was confirmed using Sequencher 5.4.6 software (Gene Codes Corporation, Ann Arbor, MI, USA). Tissue culture media was prepared as described by Nishimura et al. [27]. Scutellar calluses of rice (‘Nipponbare’ background) were generated from mature seeds and maintained on N6D media in the dark at room temperature. Two- to three-week-old cultures were bombarded with 1-μm gold particles coated with an equimolar mixture of pACL1 and pHPT plasmid (~5 μg each) using a standard protocol with a PDS 1000/He gene gun (Bio-Rad Inc., Hercules, CA, USA). pHPT contains a 35S promoter-driven hygromycin phosphotransferase gene that serves as the selection marker gene. After bombardment, the calluses were kept in the dark for 24 h and transferred to selection media, N6D supplemented with hygromycin (50 mg L^−1^). The calluses were kept on selection media, 25 °C, 16 h light, for 4 to 6 weeks. The tolerant events were transferred to a new plate (2N6D + Hygromycin) and allowed to grow until they were ready to transfer to regeneration media, as described by Nishimura et al. [27]. A total of ~20 events were transferred to the regeneration medium. Two to three weeks later, only one event successfully regenerated into plantlets. When all plantlets reached approximately 1.0–2.0 cm in length, they were transferred to a rooting medium (MS1/2 supplemented with 50 mg L^−1^ hygromycin) under 16 h light duration. One to two weeks later, plantlets of approximately 8 cm in height, with well-developed roots, were transplanted into commercial soil (Sunshine^®^ Premix No. 1; Sun Gro Horticulture, Bellevue, WA, USA) and grown to maturity. These primary transgenic plants were referred to as T_0_ plants. Subsequently, only one T_0_ plant survived to maturity. T_1_ seeds were harvested and used to investigate tolerance to herbicide.

### 2.2. Transgenic Rice Response to Foliar-Applied Fomesafen

The T_1_ rice seedlings were treated with fomesafen at the V3 stage (third-leaf collar visible [28]). The seedlings were sprayed with 780 g ha^−1^ (2×) fomesafen (Flexstar^®^, Syngenta Crop Protection, Greensboro, NC, USA) with 0.5% *v*/*v* non-ionic surfactant (Induce, Helena Chemical, Collierville, TN, USA). This dose corresponds to twice the maximum recommended dose in soybean. The herbicide was applied with a CO_2_-pressurized backpack sprayer attached to a handheld spray boom fitted with one 8002 XR even flat fan nozzle (Teejet, Wheaton, IL, USA) calibrated to deliver 187 L ha^–1^ at a pressure of 276 kPa. The plants were returned to the greenhouse after treatment and watered 48 h later and as needed. A total of 28 T_1_ plants (one plant pot^−1^) were sprayed. The experimental units were arranged in a completely randomized design. Wild-type (WT) rice plants were used as the susceptible reference. Plant injury (%) was evaluated at 2 weeks after treatment (WAT) on a rating scale of 0 to 100%, where 0 = no injury and 100 = dead plant without green tissue [29,30]. The respective nontreated checks of WT and T_1_ were used for comparison. Leaf tissues were collected from T_1_ plants. The presence of the transgene was verified by PCR as described in the cloning section. Eighteen healthy T_1_ plants, which were highly tolerant to foliar-applied fomesafen and showed the presence of transgene, were grown to generate T_2_ seeds. For further investigation, T_2_ seeds from 18 T_1_ plants were combined to create a bulk T_2_ population.

### 2.3. Transgenic Rice Response to Soil-Applied Fomesafen

Flats (12.2- by 9.5- by 5.7 cm) were filled with a 1:1 ratio of field soil and commercial potting soil. Before planting, the soil-filled flats were saturated and allowed to drain to pot water-holding capacity. Eight T_2_ (from the bulk population) or WT rice seeds were planted in each flat. Immediately after planting, the flats were sprayed with 390 g ha^−1^ fomesafen (1×). The herbicide was applied with a CO_2_-pressurized backpack sprayer attached to a handheld boom fitted with one 11002 XR even flat fan nozzle calibrated to deliver 187 L ha^–1^ at 276 kPa. The experiment was arranged in a completely randomized design with three replicates and two runs. Each flat was one replication. Nontreated checks were included. Seedling emergence count and visible injury of each emerged seedling (%) were evaluated at 3 WAT [29,30]. Height (cm), number of tillers, and number of panicles were recorded at the reproductive stage. Germination reduction (%) relative to nontreated checks was calculated using the formula:$$\mathrm{Germination}\;\mathrm{reduction}\;\left(\%\right):=\frac{\mathrm{Germination}\;\mathrm{of}\;\mathrm{nontreated}-\;\mathrm{Germination}\;\mathrm{of}\;\mathrm{treated}\;}{\mathrm{Germination}\;\mathrm{of}\;\mathrm{nontreated}}\times 100$$

Leaf tissues from nontreated T_2_ plants were also collected to verify the frequency of transgene presence (%) in the T_2_ generation and relate this to the frequency of survivors from the soil-based assay.

### 2.4. Molecular Analysis of Transgenic Rice Plants

Leaf tissues were collected from T_0_ and T_1_ plants, and genomic DNA was extracted following a modified version of the CTAB protocol [31]. The presence of *ΔG210-Apppo2* transgene was confirmed using *Apppo2* gene-specific primers and Zm*Ubi* and *nos* primer (Appendix A) using genomic DNA of the T_0_ transgenic rice line. Transgene expression was measured in selected 15 T_2_ rice plants using total RNA extracted from leaves following a modified extraction protocol developed by Hongbao et al. [32] using Trizol^®^ Reagent (Invitrogen, Carlsbad, CA, USA). A total of 1 μg RNA was used for cDNA synthesis using the qScript^®^ cDNA SuperMix kit (Quanta BioSciences, San Diego, CA, USA). The qPCR reaction was carried out using iTaq Universal SYBR^®^ Green SuperMix (BioRad, Hercules, CA, USA) using primers listed in Appendix A. The qPCR was conducted using a CFX96 Real-Time PCR machine (BioRad, Hercules, CA, USA) using the following conditions: 2 min at 95 °C, followed by 40 cycles of 30 s denaturation at 95 °C, 1 min annealing at 59 °C, 1 min extension at 65 °C, followed by melt curve analysis. Each sample was analyzed in two technical replicates. Primers were designed in such a way that they can distinguish between the target *ΔG210-Apppo2* transgene and the native *OsPPO2* gene. The Ct values were normalized against the native *OsPPO2* and housekeeping gene ubiquitin (*ubiQ*). The fold-change in gene expression was calculated using the 2^−ΔΔCt^ method [33].

### 2.5. Agar-Based Germination Assay with Transgenic Rice Line

Following the protocol by Nishimura et al. [27], rice seeds (WT and T_2_) were dehulled and then surface-sterilized. The sterilized seeds were placed in Petri plates containing 25 mL of half-strength MS medium with 0, 5, 10, 20, 40, 60, 80, and 100 μM of fomesafen. The stock solution (45,580 μM) was prepared by dissolving 50 mg technical grade fomesafen (Sigma Aldrich, St. Louis, MO, USA) in 250 μL acetone. The control plate was supplemented with just acetone to assess the potential toxicity of acetone. Each plate was divided in two; one half contained 5 seeds of T_2,_ and the other half had 5 WT seeds. The plates were incubated at 26 °C. Root growth (%) was scored visually at 2 WAT relative to a nontreated check. Leaf tissues of T_2_ survivors from 20–100 μM treatments were collected for confirmation of transgene by PCR.

### 2.6. Generation of Transgenic A. thaliana Line

The transgenic Arabidopsis line was generated under controlled conditions at BASF in Ludwigshafen, Germany. To prepare the transgene, *ppo2* containing *ΔG210* was inserted into the RTP6557 transformation vector, which was then inserted into *Agrobacterium tumefaciens* strain C58C1pMP90. Acetolactate synthase–herbicide-resistance gene was used as a selectable marker to identify transformed Arabidopsis seedlings. Plant transformation was conducted using the floral-dip method previously described [34]. After dipping, plants were kept in a cabinet under high humidity and low light intensity for 24 h and then grown under long-day conditions until maturity. The T_1_ seeds were collected and stored at 4 °C. After 15 days, T_1_ seeds were sown into GS90 soil, and 5% sand was then treated with 20 ppm imazamox to select transformed plants. These were grown under short-day (8 h light/16 h dark) for 12–14 days. Resistant seedlings (4-leaf stage) were transplanted into 6 cm pots filled with GS90 soil and grown to maturity. In the following generation, homozygous T_2_ seeds were identified.

### 2.7. Transgenic A. thaliana Response to Soil-Applied Fomesafen and Saflufenacil

Arabidopsis is highly sensitive to herbicides in comparison to rice; therefore, it is an excellent plant model to test whether the mutant *ppo2* transgene can confer resistance to PPO-inhibiting herbicides such as fomesafen and saflufenacil. For the soil-based assay, three laboratory spatula scoops of T_2_ Arabidopsis seeds were placed in falcon tubes containing 45 mL 0.1% agar and refrigerated for 3–4 days to stratify the seeds. For the dose–response assay, pots were filled with soil, watered, and seeds were sown. For sowing, 2 mL of the agar/seed mixture were used per pot which was around 50 seeds per pot. Nine rates of fomesafen (0, 0.08, 0.25, 0.74, 2.22, 6.67, 20, 60, and 120 g ha^−1^) and eight rates of saflufenacil (0, 0.25, 0.74, 2.22, 6.67, 20, 60, and 120 g ha^−1^) were applied. After application, the pots were placed into the Phytotrons and covered with hoods with 22 °C day/20 °C night—16 h day/8 h night conditions. After 5 days, the hoods were removed when seeds were germinated. The emerged seedlings were counted 20 days after planting.

### 2.8. Statistical Analysis

To obtain a clear distinction between T_1_ rice plants exhibiting high or low tolerance, injury data from foliar application of fomesafen to WT and T_1_ rice seedlings were subjected to hierarchical cluster analysis using Ward’s Minimum method in JMP Pro v.15 (SAS Institute, Cary, NC, USA). Constellation plots were produced to show the clustering patterns. Additionally, a scatter plot was prepared in SigmaPlot V.14.0 (Systat Software, San Jose, CA, USA) to visualize the correlation between transgene presence and plant injury level.

The germination reduction data obtained from the soil-based assay were subjected to analysis of variance (ANOVA) using the GLIMMIX function in SAS v. 9.4 (SAS Institute, Cary, NC, USA). The run x treatment effect was not significant; therefore, data from the two runs were combined. Since the germination reduction data did not fit a normal distribution via Shapiro–Wilk test, the β distribution was assumed for this response analysis [35]. The Student’s *t*-test (*p* < 0.05) was used to compare the means if the treatment effect was significant. Injury per survivor data were also subjected to hierarchical cluster analysis as described previously. Plant height, number of tillers, and number of panicles per plant within each injury-level cluster were subjected to ANOVA using the JMP Pro v.15 (SAS Institute, Cary, NC, USA). Phenotypic data from nontreated WT plants were used as references. When the treatment effect was significant, Fisher’s protected LSD (*p* < 0.05) was used to compare means. Gene expression was also quantified in 15 selected T_2_ survivors representing all injury level categories. The fold-change in gene expression and injury per plant were analyzed by Pearson’s correlation in JMP Pro v.15 (SAS Institute, Cary, NC, USA).

Root growth rating data from the agar-based assay were analyzed by regression using the “drc” package in R 3.5.1 [36]. A three-parameter log-logistic model was used as defined by the formula below, where Y is the root growth (%), d is the upper horizontal asymptote, x is the fomesafen dose, and b is the slope around ED_50,_ which is the dose causing 50% response level of Y [37].
$$Y\;=\frac{d}{1+\exp\left\{b\left[\log\left(x\right)-\log\left(\mathrm{ED}50\right)\right]\right\}}$$

The herbicide dose that would inhibit root growth by 50% (ED50) was estimated using this equation.

## 3. Results

### 3.1. Transgenic Rice Response to Foliar-Applied Fomesafen

A foliar application assay conducted with plants derived from PCR-verified T_0_ plants (Appendix A) showed that none of the T_1_ plants were controlled 100% with foliar-applied fomesafen. The foliar injury ranged from 0 to 60% among T_1_ and from 30 to 60% among WT plants. Thus, some transgenic plants were as sensitive as the WT plants. The hierarchical cluster analysis revealed two distinct groups of plants based on injury levels (Figure 1a). Cluster 1 was characterized as “highly tolerant” and cluster 2 as “minimally tolerant”. Cluster 1 consisted of plants with <10% injury and was comprised only of T_1_ plants. Sixty-eight percent (68%) of T_1_ plants were highly tolerant to fomesafen, showing low injury. All T_1_ plants with low injury harbored the transgene. Plants in cluster 2 exhibited 30 to 60% injury and included T_1_ and WT individuals. Out of 28 treated T_1_ plants, 4 tested negative for the transgene (Figure 1b). T_1_ plants without the transgene fell in cluster 2, among the highly injured individuals. Despite the strong correlation between transgene presence and low injury, five transgene-positive T_1_ plants showed high injury and grouped with cluster 2.

### 3.2. Transgenic Rice Response to Soil-Applied Fomesafen

Soil-applied fomesafen delayed rice emergence for at least 5 days (up to one week) (Figure 2). The germination of WT seeds was reduced by 92% by fomesafen, and that of T_2_ seeds was reduced by 27% (Figure 2). The WT seedlings (8%) did not grow further and died, whereas T_2_ seedlings (73%) survived. All T_2_ survivors showed the presence of the transgene and were classified as tolerant. From 33 nontreated plants, 82% harbored the transgene [38]. Considering that 73% of T_2_ plants survived, the presence of the transgene can be correlated with tolerance to the pre-emergence application of fomesafen. Injury levels among T_2_ survivors varied widely (from 30 to 95%) and were grouped in three well-defined clusters. T_2_ survivors with 30–50%, 60–70%, and 80–95% injury were classified as highly tolerant, moderately tolerant, and slightly tolerant, respectively (Appendix A). The rating of injury levels in T_2_ plants included delayed emergence, stunting, and foliar damage compared to nontreated plants. Although T_2_ plants in moderately tolerant and slightly tolerant clusters still incurred high injury, all T_2_ plants that survived fomesafen were transplanted and survived. The phenotypic traits (height, number of tillers, and number of panicles) of plants across clusters and WT plants were subjected to ANOVA. The plants in cluster 1 (<50% injury) were phenotypically different from those in cluster 3 (>80% injury) (Figure 3). Overall, plants in cluster 1 were the tallest across clusters, with the highest number of panicles. Plants in cluster 3 incurred substantial injury, which delayed phenological development. Despite some initial injury, plants in cluster 1 had significantly more panicles than the nontreated WT plants.

### 3.3. Gene Expression Analysis of Fomesafen Tolerant Rice Plants

The T_2_ plants that survived soil-applied fomesafen showed a wide range of injuries. To understand this phenotypic variation, expression analysis of *ΔG210*-*Apppo2* transgene was performed. Quantitative PCR analysis showed that all T_2_ survivors carrying the transgene had > 150-fold expression of the transgene compared to WT. The majority (10 of 15) of T_2_ plants showed transgene expression between 150 to 400-fold when calculated against native *OsPPO2* (Figure 4). With one exception, individuals showing low injury had transgene expression values < 400-fold. When the transgene expression was calculated against ubiquitin, the majority of T_2_ survivors (12 out of 15) showed transgene expression < 800-fold (Appendix A). The injury per survivor did not correlate with fold-change in transgene expression calculated against *O. sativa PPO2* (*r* = 0.3411, *p* = 0.2134) or rice ubiquitin (*r* = 0.1044, *p* = 0.7112).

### 3.4. Agar-Based Germination Assay with T_2_ Seeds of Transgenic Rice

To further characterize the pre-emergence tolerance conferred by the *ΔG210*-*Apppo2* transgene, an agar-based germination assay was conducted using the T_2_ bulk population. All T_2_ and WT seeds germinated in agar media supplemented with up to 100 µM fomesafen. However, the root growth of WT seedlings was inhibited even at the lowest concentration (5 μM) of fomesafen (Figure 5). Therefore, only the T_2_ root growth rating data fitted the regression model (Appendix A). Root growth of the majority of T_2_ seedlings (88%) was not inhibited with up to 10 μM fomesafen but was reduced to 50% at 20 μM. Although reduced, roots continued to grow in all five T_2_ seedlings plated in MS media supplemented with 20, 40, and 60 μM fomesafen, whereas root growth of WT seedlings in the same treatments was negligible. Only one of five T_2_ seedlings had root growth on media supplemented with 80 and 100 μM fomesafen. The estimated fomesafen concentration that would result in a 50% growth reduction in the T_2_ population was 45 μM (Appendix A). Except for one seedling (that germinated on 60 μM fomesafen), *ΔG210-Apppo2* transgene was present in all T_2_ seedlings that exhibited root growth.

### 3.5. Transgenic A. thaliana Response to Soil-Applied Fomesafen and Saflufenacil

We generated a transgenic *A. thaliana* line using the *ΔG210-Apppo2* gene to demonstrate whether *ΔG210-Apppo2* can confer high tolerance to pre-emergence PPO inhibitors in the highly sensitive Arabidopsis. In the dose–response assay, the transgenic *A. thaliana* line was more tolerant to soil-applied herbicides than WT plants. WT plants were killed 100% at 2.22 and 6.67 g ha^−1^ fomesafen and saflufenacil, respectively (Figure 6 and Appendix A). On the other hand, the transgenic line was not completely controlled at the highest rate (120 g ha^−1^) of both herbicides. The dose–response experiment showed that the transgenic *A. thaliana* line was tolerant to both PPO herbicides tested, whereas WT plants were highly susceptible.

## 4. Discussion

Resistance to herbicides is one of the most important traits utilized in plant biotechnology, which is widely used to improve agricultural efficiency. There are more than 20 gene products targeted for herbicide action. The PPO is an ideal target as it is highly conserved and has a critical function within the plant. Field-evolved PPO mutations discovered in PPO-inhibitor-resistant weeds can be a tool for future crop engineering research. According to Kim et al. [39], G399A mutation is deemed the most appropriate or suitable site for gene-editing to make PPO-inhibitor resistant tomatoes. Recently, a mutant *ppo2* gene having R128G mutation was isolated from PPO-resistant *Amaranthus retroflexus* L (*Arppo2*) and overexpressed in Arabidopsis. The presence of this mutation induced high tolerance to the foliar-applied PPO-herbicides (fomesafen, lactofen and carfentrazone-ethyl) tested [25].

In our study, we overexpressed the *A. palmeri ppo2* gene containing field-evolved *ΔG210* mutation in rice (as a crop model). Rice has some level of tolerance to foliar-applied PPO herbicides but not enough for commercial application of fomesafen. Our data on WT plants reflected this. In preliminary dose–response experiments conducted with the WT biotype, WT seedlings were not killed with rates up to eight times the maximum fomesafen dose [38]. Other diphenylether herbicides, which are analogs of fomesafen (chlomethoxyfen and oxyfluorfen), are safe to use on rice postemergence [40,41]. Even though none of the rice plants (WT or transgenic T_1_) were killed by foliar-applied fomesafen, all T_1_ plants that harbored the *ΔG210-Apppo2* transgene showed low injury, indicating a reduced phytotoxic effect of fomesafen. To confirm the benefit of carrying *ΔG210-Apppo2* transgene, it is necessary to have treatments that would obtain 100% control. This was achieved with the pre-emergence application of fomesafen to transgenic rice.

The soil-based assay confirmed that *ΔG210-Apppo2* transgenic rice has a high tolerance to soil-applied fomesafen. The tolerance to soil-applied fomesafen in transgenic rice may be due to the constitutive transgene expression being driven by the maize ubiquitin promoter, which is active in germinating seedlings. Maize ubiquitin promoter directs high transgene expression in many young rice tissues [42,43], affecting the high expression of the transgene in rice seedlings. Tissues of germinating seeds and newly germinated seedlings are not equipped to counteract the strong oxidative effects of fomesafen action. By overexpressing a tolerant *Apppo2*, we hypothesize that the oxidative stress would be reduced, and sufficient functional protein would allow seedlings to continue growing.

Regarding the transgene effect on agronomic traits, the T_2_ survivors that had the lowest injury from the soil-based assay had a significantly higher number of panicles than the nontreated WT rice plants. This result suggests that overexpression of *ΔG210-Apppo2* in rice may affect yield. To test this hypothesis, field experiments are needed to assess the transgene effect on yield and yield components, such as the number of seeds per panicle and the weight of 1000 seeds. Manipulation of PPO expression can produce divergent results. Jung et al. [44] attempted to increase the mitochondrial PPO activity by overexpressing human PPO in rice and instead increased the PPO activity in the chloroplast, which resulted in severe necrosis and growth inhibition. Yet, in another study, the advanced generation of the transgenic rice line (M4) expressing *Mx*PPO produced a higher yield compared to WT when treated with PPO herbicides [45]. Lately, the oxyfluorfen-resistant rice line (M-206), which was developed in California, contains the mutant allele ROXY that confers non-transgenic resistance to this herbicide. Field testing with seeds from the ninth generation of this rice line showed a significantly higher yield than M-206 without the ROXY mutant allele after oxyfluorfen applications [46]. In these studies, the yield increase in comparison to WT is due to the combination of high WT injury and effective weed control by the PPO herbicides; there was no evidence that PPO manipulation directly contributed to the improvement of yield potential. Even though transgenic herbicide-resistant rice would be beneficial and profitable to rice farmers, it is unlikely that any transgenic rice varieties will be commercialized soon. Consumer acceptance is a major constraint to transgenic rice production and commercialization [47,48].

Initially, we hypothesized that the expression level of *ΔG210-Apppo2* would be positively correlated with fomesafen tolerance. Our data did not support this hypothesis. Even though the transgene presence provided tolerance to foliar- and soil-applied fomesafen, plants displayed variable tolerance to fomesafen, which could be due to varying levels of transgene expression with the presence of hemizygous vs. homozygous states of the transgene. In PPO-resistant *A. palmeri*, Carvalho-Moore et al. [49] showed that among the resistant accessions tested, the accession with a high frequency of survivors that are homozygous for *Δ**G210* also had higher ED50 and less injury at a high dose of fomesafen. Variation in transgene expression is expected in the progeny since transgenic individuals generally differ in transgene expression, and full or partial levels of gene expression may be heritable [50]. Gene silencing factors and gene segregation across generations can also affect transgene expression in the offspring [51,52,53] until such a trait is fixed in subsequent generations.

An agar-based assay is an effective and quick technique to characterize plant response to herbicides. In our experiment, the root growth of WT rice seedlings was inhibited even at the lowest concentration of fomesafen. In a similar study, Lee et al. [54] showed that transgenic rice line overexpressing human PPO was tolerant to up to 5 μM of oxyfluorfen in agar-based seed germination assay, while the wild-type was sensitive to the lowest concentration of 2 μM. Like fomesafen, oxyfluorfen is also a diphenylether PPO herbicide. A transgenic rice line expressing *Myxococcus xanthus* PPO (*Mx*PPO) showed high tolerance to oxyfluorfen, whereas the wild-type did not germinate at the lowest concentration tested [55].

Rice has natural tolerance to foliar-applied fomesafen due to higher antioxidative activities [56]; therefore, *A. thaliana,* which is a very sensitive species to PPO-inhibiting herbicides, was the perfect plant model to assess the effect of overexpressing *ΔG210-Apppo2*. Recently, Huang et al. [25] also used transgenic *A. thaliana* to evaluate the effect of a mutant Arppo2 enzyme on Arabidopsis response to foliar-applied fomesafen compared to a line overexpressing the wild type ArPPO2. The former showed tolerance to fomesafen, while the latter was as sensitive as the non-transgenic Arabidopsis line. Thus, our data suggest that the overproduction of the mutant Apppo2 protein, which is not inhibited by PPO herbicides, allows normal-rate conversion of the phototoxic protoporphyrinogen IX to porphyrin (in the presence of the inhibitor herbicide) enabling the plant to survive. It would be very interesting to know if transgenic *ΔG210*-Apppo2 protein was targeted towards both chloroplast and mitochondria, to overcome the soil-herbicide effect at the germinating stage. Transgenic expression of *Apppo2* without the dual targeting signal peptide may show if simultaneous expression occurs in both organelles and if it is imperative for survival with soil-applied PPO herbicides.

In summary, the insertion of *A. palmeri ppo2* containing *ΔG210* mutation confers tolerance to fomesafen in rice and Arabidopsis. The majority of transgenic rice plants are able to fully recover from pre-emergence treatment with fomesafen, whereas WT plants are killed. Our research shows that if the increase in copy number or differential promotor activity of the *PPO* gene is selected under repetitive herbicide usage on key weed species such as *A. tuberculatus* and *A. palmeri*, the efficacy of pre-emergence PPO herbicide can be reduced. Additionally, this research presents an example of harnessing herbicide-resistant genes from weeds to develop herbicide-tolerant crops. Cross-tolerance to other soil-applied PPO-inhibiting herbicides will be determined for the *ΔG210-Apppo2* transgenic rice in future work.

## Figures and Tables

**Figure 1 genes-13-01044-f001:**
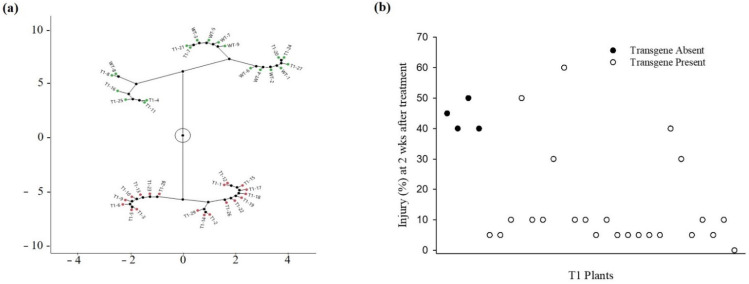
Response of transgenic rice plants to foliar-applied fomesafen (780 g ha^−1^). (**a**) Constellation plot from the hierarchical clustering analysis of T_1_ and wild type (WT) injury data collected 2 weeks after foliar treatment. Cluster 1 (red) is composed of plants showing low injury. Only T_1_ plants were part of cluster 1. Cluster 2 (green) is composed of plants showing high injury. Both WT and T_1_ were present in cluster 2. (**b**) Scatter plot of foliar injury levels of T_1_ plants with (white circles) or without (black circles) the *ΔG210-Apppo2* transgene.

**Figure 2 genes-13-01044-f002:**
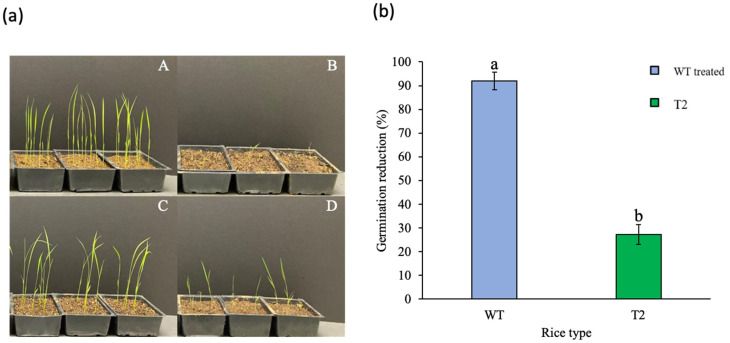
Response of transgenic rice plants to soil-applied fomesafen (390 g ha^−1^). (**a**) The response of wild type and T_2_ rice plants 3 weeks after soil treatment. The photos show one run of the experiment; (**A**) WT nontreated, (**B**) WT treated, (**C**) T_2_ nontreated, and (**D**) T2 treated. (**b**) Wild-type and T_2_ germination reduction (%) calculated 3 weeks after soil-applied fomesafen treatment. Means were derived from a combined analysis of two runs since plant response to treatments did not vary across runs. Lowercase letters indicates significant difference (*p* < 0.05).

**Figure 3 genes-13-01044-f003:**
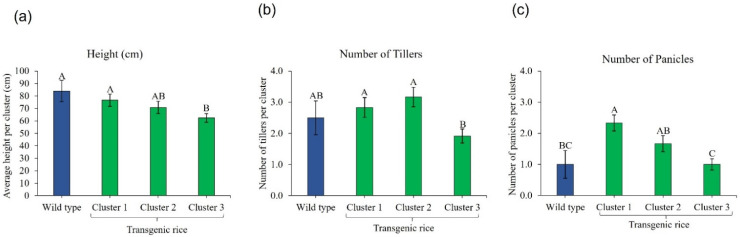
Phenotypic data of T_2_ rice plants that survived 390 g ai ha^−1^ soil-applied fomesafen. (**a**) Height, (**b**) number of tillers, and (**c**) number of panicles of T_2_ survivors by phenotypic trait cluster. Data were collected when the majority of survivors transitioned to the reproductive stage. Significant means were separated using Fisher’s protected LSD (*p* < 0.05). Means with different letters above bars are different (*p* < 0.05).

**Figure 4 genes-13-01044-f004:**
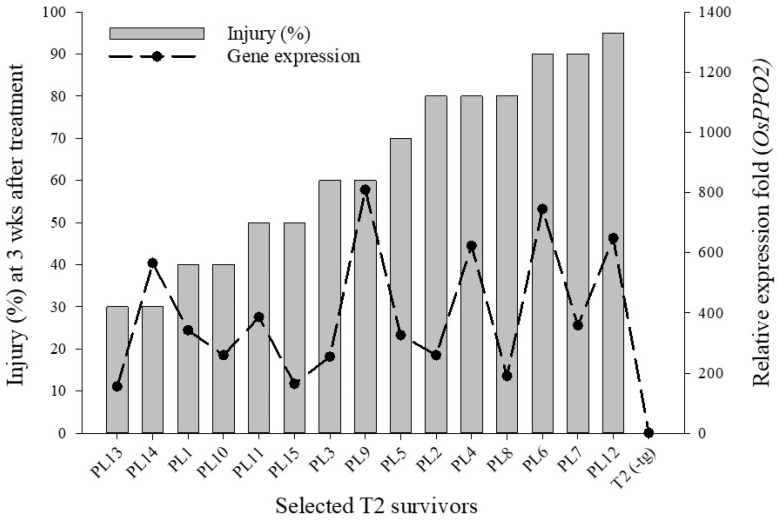
Visible injury (%) and transgene expression in T_2_ survivors in response to soil-applied fomesafen. Expression of transgene is calculated relative to native *PPO2* from *O. sativa*. Nontreated T_2_(-tg) was used as a control for transgene expression.

**Figure 5 genes-13-01044-f005:**
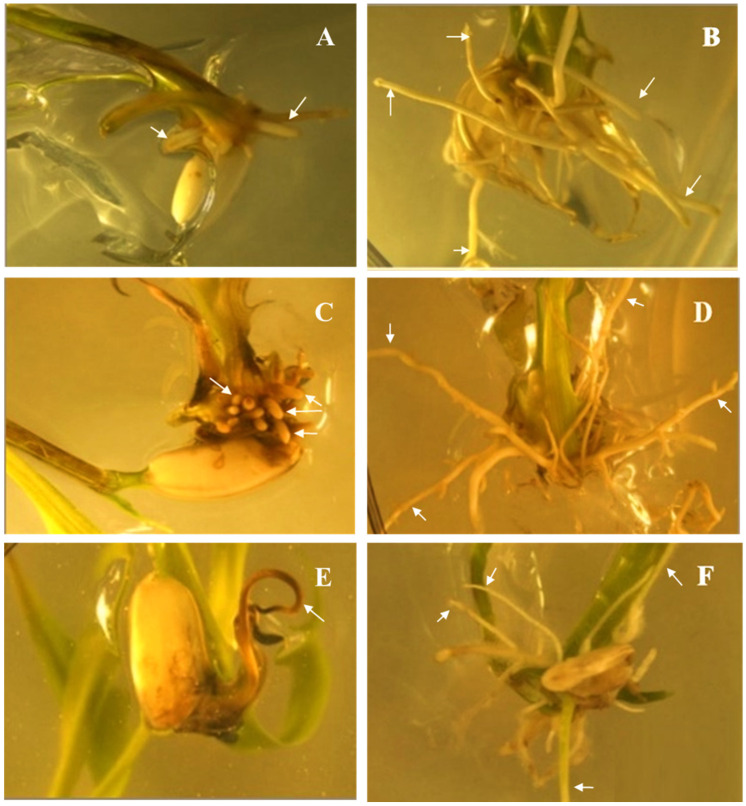
Agar-based assay showing root growth of rice seedlings in different concentrations of fomesafen. Root growth of wild type at 5 μM (**A**), 40 μM (**C**), and 100 μM (**E**), root growth of T_2_ plants at 5 μM (**B**), 40 μM (**D**), and 100 μM (**F**). White arrows show inhibited root growth in (**A**,**C**,**E**); and healthy root growth in (**B**,**D**,**F**).

**Figure 6 genes-13-01044-f006:**
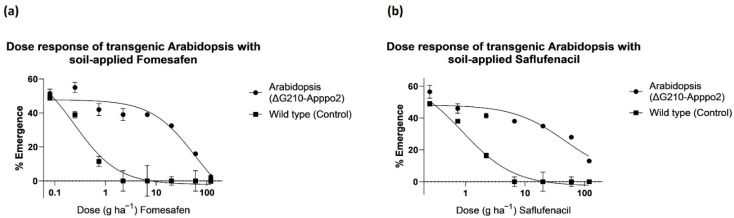
The dose–response curves for percentages of germinated transgenic (*ΔG210-Apppo2*) Arabidopsis. (**a**) Response to increasing doses of fomesafen and (**b**) saflufenacil.

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
