# Peer review of "Field-Evolved ΔG210-ppo2 from Palmer Amaranth Confers Pre-emergence Tolerance to PPO-Inhibitors in Rice and Arabidopsis"

_genes, 2022, doi:10.3390/genes13061044_

Round 1
Author Response
Response to Reviewer Comments
First of all, we thank the reviewer for the comments and for reviewing thoroughly the manuscript.
Reviewer 1
Broad comments
- This paper describes the acquisition of herbicide resistance in rice and Arabidopsis by expressing an herbicide resistant PPO allele from Amaranthus palmeri. The data presented convincingly argue for increased resistance to fomesafen in rice and to fomesafen and saflufenacil in Arabidopsis. However, I was not sure what the aim of the paper is and found the narrative extremely confusing. For example, the introduction proposes to characterize the strength of the G210 deletion for resistance using heterologous expression systems. By the end of the discussion, the paper has shifted to demonstrating “an example of harnessing herbicide-resistant genes from weeds to develop herbicide-tolerant crops.” A second example is that the title refers only to fomesafen tolerance in rice, but both rice and Arabidopsis were tested.
Response: We changed the order of statements in the conclusion since our main focus was to characterize this mutation using foreign systems to determine the level of tolerance provided by its presence. We believe that the statement “an example of harnessing herbicide-resistant genes from weeds to develop herbicide-tolerant crops” provides another example of how this research might be useful in the future without shifting its objective. Also, we have revised the title of the paper to “Field-evolved ΔG210-ppo2 from Palmer amaranth confers preemergence tolerance to PPO-inhibitors in rice and Arabidopsis”
- A major methodological issue is that only one T0 transgenic rice plant survived.
Response: We understand that use of single transgenic line is a weakness of the study; however, by including T1 segregating population, we have clearly addressed that the fomesafen tolerance is a transgene associated trait, which is the core aspect of this work. The transgene-positive and transgene-negative (null segregant) lineages showed differences in tolerance in foliar application assay (L. 276-290 and Figure 1b). Moreover, all T2 survivors of soil application carried the transgene (L. 306-307). Both herbicide assays technically showed that fomesafen tolerance in T1 and T2 lines is associated with the transgene. Therefore, by using a single T0 line, we established that PPO allele from Amaranthus palmeri confers tolerance to fomesafen.
- I found the organization of the methods confusing and propose moving section 2.5 (molecular analysis of transgenic rice plants) to after section 2.2. I also think presenting the assays by species and then by treatment method makes more sense than doing it by treatment and then by species, especially since the treatments don’t exactly match across species (e.g., saflufenacil was not used on rice). Furthermore, I expected some information about molecular analysis of transgenic Arabidopsis. The results do not need to be described in the same order as the methods but the current order, which does follow the methods, suffers from the same issues described above. Throughout the paper I found it hard to keep track of which species was being discussed.
Response: We agree that separating by species is easier to follow; therefore, the methodology and results sections were rearranged.
- Since results and discussion are separate sections, please be careful to keep discussion points out of the results section. For example, lines 309-311, “Considering that 73% of T2 plants survived, we can conclude that the presence of transgene is correlated with tolerance to preemergence application of fomesafen.” Or lines 322-324, “Despite some initial injury, plants in cluster 1 had significantly more panicles than the nontreated WT plants, indicating better health of this group of T2 plants.”
Response: These statements were revised in the manuscript (L. 310-312; L. 323-324).
- Creating and characterizing these transgenic lines is great work. I get lost, though, in why the particular characterizations were chosen, and what hypotheses were driving this work. If the experiments were more clearly justified and if the conclusions aligned with issues raised in the introduction this would be a very solid paper.
Response: Fomesafen can be used preplant, preemergence, over-the-top, or postemergence-directed in a few crops, but it is not labeled in rice. We decided to use soil and agar-based assays in the later generation (T2) since rice is more sensitive to soil-applied fomesafen (L. 67 - 68) than when it is foliar-applied. We also slightly modified the conclusion to emphasize our main goal which was to characterize the “major target site mutation to confer resistance to PPO-inhibitors” (L. 87) using foreign systems to determine the level of tolerance it provides.
Further points to consider:
- I imagine that characterizing the effect of the Gly210 deletion of the ApPPO2 gene using heterologous gene expression would require comparing two transgenic lines: one with the Wt ApPPO2 gene, and one with the ApPPO2 gene lacking Gly210.
Response: We agree that comparison of ΔG210-Apppo2 gene vs WT ApPPO2 would be the ideal case, however in this study we generated only ΔG210-Apppo2 using rice and Arabidopsis. However, our data is supported by a recent paper where other researchers have created WT PPO2 transgenic Arabidopsis line (L. 448-459).
- Are there other, possibly significant, differences in the ApPPO2 vs OsPPO2 vs AtPPO2 protein? Would deleting Gly210 from the native Arabidopsis or rice PPO2 protein confer resistance, or is there something particular to the ApPPO2 version that gives resistance? What if a PPO promoter (either from palmer or the model species) is used to drive expression instead of the ubiquitin promoter?
Response: PPO2 protein is a highly conserved protein, but glycine at 210th position in A. palmeri PPO2 is not conserved. The corresponding ApPPO2 G210 position in OsPPO2 is glycine, but AtPPO2 has alanine. Deletion of G210 in A. palmeri results from the deletion of three bp that are repeated. In other words, it is a deletion of a short repeat. Although, G210 is not conserved in Arabidopsis, it is possible that deleting a repeat at that position in native PPO2 gives tolerance to PPO-inhibiting herbicides; however, only experimental validation can prove it. We agree that these questions are extremely relevant for a future study. However, in the present paper, our main goal was to characterize the G210 deletion mutation from A. palmeri by transgenic approach. Maize ubiquitin promoter was used due to its success rate in rice. The PPO2 transformation in rice using the ballistic method was quite challenging, with a low rate of success and survival (as mentioned in the paper). Therefore, we did not have the resources to experiment with different promoters.
- Does PPO2 in rice or Arabidopsis have dual-targeting?
Response: Thus far, there is no evidence that PPO2 in rice or A. thaliana have dual-targeting. In fact, only few species exhibit dual-targeting (Dayan FE, Barker A, Tranel PJ: Origins and structure of chloroplastic and mitochondrial plant protoporphyrinogen oxidases: implications for the evolution of herbicide resistance. Pest Manag Sci 2018, 74(10):2226-2234.).
- What would make transgene expression increase several hundred fold over ubiquitin expression if transgene expression is controlled by the ubiquitin promoter? What does expression have to do with resistance when the Gly210 deletion is associated with changing the interaction of fomesafen with its binding site?
Response: We used maize promoter in the transgene construct, while gene expression was measured against rice ubiquitin gene. We have added more information in the revised manuscript (L. 119, 189, 420-421) to clarify the origin of the promoter. Overexpression of a foreign gene ensures that there would be viable protein production and if it has a role it can manifest the phenotype, which in our case, is fomesafen tolerance.
- Was transgene copy number tested at any point in rice? Copy number variation may explain the variable responses to the herbicide.
We have not clearly determined the copy number in the transgenic rice. Quantitative PCR assay was done to estimate the relative copy number in T2 plants and we found no correlation of copy number with the phenotypic responses to the herbicide or gene expression, however optimization using different concentration of gDNA was not performed. Therefore, we decided to not include this data since it was not very informative.
Specific comments
- Abstract: “Foliar assay was conducted with twice the labeled dose of fomesafen in soybean (780 g ha-1) using T1 plants of transgenic rice.” It isn’t clear why the fomesafen dosage is expressed in terms of the soybean label when the chemical is being applied to rice carrying an A. palmeri gene.
Response: Fomesafen is labeled and highly used in soybean in Arkansas (Commercial label:http://www.cdms.net/ldat/ld6BM003.pdf; MP44 for Recommended Chemicals in Weeds and Brush control: https://www.uaex.uada.edu/publications/pdf/mp44/mp44.pdf ). It is not labeled in rice (L. 72 – 73).
- I was surprised by saflufenacil and recommend mentioning it earlier in the abstract.
Response: Revised in the manuscript.
- Line 87: I don’t know what ‘TSM’ stands for.
Response: Revised in the manuscript.
- Line 202: ‘spatulas’?
Response: We have revised the description in L. 235.
- Line 204: I expected the soil source to be described similarly to the rice experiments.
Response: Soil source has been described in the citation we have used for floral-dip method: Arabidopsis thaliana seeds (stock MC24, from the Max Planck Institute for Molecular Plant Physiology at Golm) were sown into a substrate composed of GS90 soil + 5% sand.
- Lines 202, 204-205: Perhaps the number of seeds used per assay could be condensed into a single statement.
Response: Revised in the manuscript.
- Lines 206-208: Rice was sprayed with one rate, which was justified based on the labeled dose for soybean. I think it would be helpful to briefly mention why a range of rates was used on Arabidopsis, why two PPO inhibitors were tested, and why the particular rates were chosen.
Response: Revised in the manuscript.
- Section 2.5: Please briefly describe the method for genomic DNA extraction. Also please include more detail about the leaves used to measure transgene expression.
Response: Revised in the manuscript.
- Lines 243-244: Please specify which T1 plants: rice, Arabidopsis or both
Response: Revised in the manuscript.
- Lines 246-248: I found this to be extraneous information.
Response: We agree that this information might seem unnecessary. However, we believe it is important to provide all information regarding analysis and software used in this study for future readers.
- Lines 258-259: ‘Phenotypic data from nontreated WT plants were used as references’- were nontreated transgenic plants phenotypically distinct from WT?
Response: Nontreated checks for T2 and WT were included in the assay (L.173-174), and we did not observe any distinctions between the biotypes.
- Line 260: ‘Gene expression’: to me this implies an RNA-seq experiment was performed, but I suspect you are referring to the genes selected for qRT-PCR. In that case, I recommend explicitly referring to that section here.
Response: Although RNA-seq is one excellent method to obtain gene expression, this is not the only method available. qRT-PCR is a standard methodology to access gene expression.
Please refer to the following links for previous examples with gene expression obtained by qRT-PCR in weed science and biotechnology, respectively: 1) Gaines et al. (2010). Gene amplification confers glyphosate resistance in Amaranthus palmeri (https://doi.org/10.1073/pnas.0906649107); 2) Nguyen et al. (2014). Strong activity of FLPe recombinase in rice plants does not correlate with the transmission of the recombined locus to the progeny (https://doi.org/10.1007/s11816-014-0332-5).
- Line 266: ‘the formula below’ It does not seem that the formula was included.
Response: The formula was included in the manuscript (L. 271).
- Lines 305-309: I do not understand the connection between 1) the frequency of the transgene in the T2 population and 2) if the presence of the transgene correlates with fomesafen tolerance in T2 plants.
Response: After evaluation in soil-applied herbicide assays, we were not able to collect tissue from non-survivors T2 seedlings as they die or don’t grow. Therefore, to know the frequency of transgene in T2 population we used nontreated T2 seedlings. Out of nontreated T2 seedlings, 82% carried the transgene, and in soil-assay, 73% T2 seeds germinated and survived in the presence of fomesafen (all carrying the transgene L. 306). We interpret that tolerance to fomesafen in T2 seedlings were due to presence of transgene because the frequency of transgene carrier in T2 population is comparable to the frequency of soil-survivors,
- Lines 311-312: how were the groups identified? Were there clusters among WT plants?
Response: As mentioned in L. 305-306, the few WT seedlings that germinated did not survive after transplanting. Therefore, the clusters were formed only with T2 plants. The clusters identification was provided L. 312-315 and table S2.
- Lines 309, 317: In line 309, 73% of T2 plants survived but in line 317, all T2 plants survived, and I believe plants from the same experiment are being described. The group of plants whose survival is of interest needs to be distinguished more clearly, or the different benchmarks for survival need to be clarified.
Response: We meant that all T2 survivors transplanted survived. This was clarified in the paper.
- Figure 2A – check that the panels are labeled correctly. I assumed C was T2 nontreated and D was T2 treated, but the legend indicated the opposite.
Response: Revised in the manuscript.
- Figure 2B – The figure shows germination reduction (%) but the legend says germination (%).
Response: Revised in the manuscript.
- Figure 3 – connecting letters are referred to in the legend but do not appear in the figure. Panels b and c should include ‘average’ on the y-axis label.
Response: Revised in the manuscript.
- Figure 4 – do not include -10 on the y-axis. I have never seen a dose-response curve with the highest dose on the left - usually the x and y axes intersect at 0. In fact, the figure does not present a dose-response curve, which would be the output from fitting a model using drc (as in Figure S4) and instead just ‘connects the dots’.
Response: Dose-response graph is added.
- Figure 6 – it took me longer than I wanted to understand the root phenotype I was intended to see. Could these images be cleaned up at all? I recommend at minimum indicating a scale and cropping out the sharpie marks. But adjusting brightness/contrast, or adding some marker to make the (non)roots obvious, would be really great.
Response: This is now figure 5. It is revised according to suggestions.
- Figure S2b- please redo this so the sequences are presented in a standard font size. The screenshot is too small for me to read easily.
Response: Revised in the supplementary material.
- Figure S3 – the left y-axis needs a label
Response: Revised in the supplementary material.
- Table S2 – Perhaps there is a more intuitive way to present this information.
Response: We think it is the best way to present it as a supplementary information.
- Tables S3,S4 – As with Figure 4, I find it very confusing to have the doses presented from highest to lowest.
Response: Revised in the manuscript.
Reviewer 2 Report
Please see my comments in the attached file. Thanks

Author Response
Response to Reviewer Comments
First of all, we thank the reviewer for the comments and for reviewing thoroughly the manuscript.
Reviewer 2
Line 61 – 62: What is the biological significance of the compartmentalization.
Response: included in the manuscript (L. 62-63).
Line 163: Why soybean rate was chosen and how much do they vary for other crops?
Response: The commercial fomesafen formulation (Flexstar) used in this work is only labeled in soybean applications. We used the herbicide label (http://www.cdms.net/ldat/ld6BM003.pdf) and the MP44 – Recommended Chemicals for weed and brush control to choose the dose (https://www.uaex.uada.edu/publications/pdf/mp44/mp44.pdf).
Line 183: Why T2 were used? Do authors performed the same experiments with T1 lines?
Response: Only T2 seeds were used for the soil-applied and agar-based assay. We decided to conduct these assays only with the second generation due to the injury variability observed on foliar assay with T1. Also, T1 seedlings were originated from only one T0 plant, and seed supply was limited.
Figure 3: Letters are missing in the graph.
Response: Revised in the manuscript.
Figure 3: It looks like wild type have higher height than transgenic rice, what could be the reason? Is there any fitness cost related to transgene?
Response: Soil-applied fomesafen delayed rice emergence for at least 5 d (up to one week) (L. 300). The wild type in Figure 3 is nontreated with fomesafen; therefore, these plants had no delay in germination and were ahead physiologically compared to the treated ones. We did not observe fitness cost related to transgene between nontreated WT and T2.